# Dietary Acrylamide Induces Depression via *SIRT3*-Mediated Mitochondrial Oxidative Injury: Evidence from Multi-Omics and Mendelian Randomization

**DOI:** 10.3390/cimb47100836

**Published:** 2025-10-10

**Authors:** Lele Zhang, Shun Li, Shengjie Liu, Zhenjie Wang

**Affiliations:** 1School of Biological and Food Engineering, Fuyang Normal University, 100 Qinghe West Road, Fuyang 236041, China; zhanglele@fynu.edu.cn (L.Z.); 23211304@stu.fynu.edu.cn (S.L.); 2College of Food Science and Technology, Nanjing Agricultural University, No. 1 Weigang Road, Nanjing 210095, China

**Keywords:** acrylamide, major depressive disorder, mitochondrial oxidative stress, *SIRT3*, environmental toxicant, molecular docking

## Abstract

Acrylamide (ACR), a common dietary pro-oxidant generated in heat-processed foods, disrupts mitochondrial redox homeostasis. While its neurotoxic effects are recognized, the role of ACR in depression remains poorly understood. We hypothesized that dietary ACR exposure promotes depression via SIRT3-dependent mitochondrial oxidative injury. Through an integrative approach combining network toxicology (to prioritize candidate targets), transcriptomics, and Mendelian randomization (MR), we identified SIRT3 as the central mediator. Molecular dynamics simulations demonstrated that ACR’s primary metabolite glycidamide (GA) formed more stable and rigid complexes with key targets (including SIRT3, TP53, CASP3, JUN, PTGS2, and PTK2) than ACR itself, as evidenced by superior structural stability, reduced flexibility, and enhanced hydrogen bonding. Transcriptomic analysis of the human prefrontal cortex (datasets GSE54567 and GSE54568) revealed mitochondrial deacetylase sirtuin 3 (*SIRT3)* as the most significantly suppressed gene in depression (*p* < 0.01), suggesting an impairment in Superoxide dismutase 2 (SOD2)-mediated antioxidant defense. MR further established *JUN* and *PTK2* as causal genetic risk factors for depression (*JUN*: Odds Ratio (OR) = 1.029, 95% CI = 1.002–1.057; *PTK2*: OR = 1.040, 95% CI = 1.005–1.076; *JUN* (OR) = 1.048, 95% CI = 1.021–1.076, *PTK2*: OR = 1.073, 95% CI = 1.039–1.109) of each MR estimates, while other candidates lacked genetic support. Our findings demonstrate that ACR induces depression primarily through SIRT3 suppression, activating JUN/PTK2 pathways, suggesting its potential role in environmental toxicant-induced redox imbalance.

## 1. Introduction

The Maillard reaction, a non-enzymatic process between amino acids and reducing sugars during high-temperature processing, generates a range of Maillard reaction products (MRPs) in thermally processed foods, including fried snacks, baked goods, and roasted coffee [1]. This ubiquitous reaction is valued in the food industry for enhancing color, aroma, and flavor. It occurs not only in food matrices but also in biological systems and soil environments. While certain MRPs exhibit beneficial properties (antioxidant, antimicrobial), others act as pro-oxidants [2,3] or cytotoxic agents. Among these, acrylamide (ACR), discovered in food in 2002, has been recognized as a potent food-borne toxicant with carcinogenic and neurotoxic potential [4].

Upon ingestion, ACR is rapidly absorbed and systemically distributed, showing particular accumulation in nervous tissues (e.g., brain, spinal cord) and metabolic organs (e.g., liver) [5]. Its metabolism proceeds primarily through two competing pathways: primary conjugation with glutathione (GSH) leading to urinary excretion [6], and CYP2E1-mediated oxidation forming glycidamide (GA)—a genotoxic epoxide [7]. Like ACR, GA remains electrophilically reactive and is implicated in ACR-induced toxicity through its ability to form adducts with biomolecules. Preclinical studies confirm its multi-organ toxicity, including neurotoxicity, genotoxicity, and reproductive dysfunction [8]. The neurotoxicity of ACR primarily arises from the covalent adduction of free thiol groups in critical antioxidants (e.g., glutathione, thioredoxin), thereby depleting cellular redox defenses and inducing endoplasmic reticulum stress, which activates apoptotic pathways [9]. For instance, Ma et al. demonstrated that ACR exposure elevates reactive oxygen species (ROS), superoxide anion (O_2_^−^), and hydrogen peroxide (H_2_O_2_) levels while concurrently depleting glutathione (GSH)—a redox impairment phenomenon further validated in C. elegans models [10]. Notably, such mitochondrial dysfunction and oxidative neuronal damage represent a well-established pathomechanistic substrate underlying neuropsychiatric disorders. We posit that this damage is directly mediated by GA-induced suppression of the mitochondrial deacetylase sirtuin 3 (SIRT3). For instance, Li et al. demonstrated that ACR exposure significantly reduces NAD^+^ levels and downregulates the expression of SIRT3 in the liver and hepatocytes [11]. Given that SIRT3 is a master regulator of mitochondrial antioxidant defenses and energy metabolism, its suppression provides a plausible mechanistic link connecting ACR exposure to the mitochondrial oxidative damage observed in the brain, which is a hallmark of depression pathophysiology.

Human exposure to ACR is widespread and primarily dietary. The Joint Expert Committee on Food Additives (JECFA) identifies major dietary sources as potato crisps, potato chips, coffee, and bread [8]. Alarmingly, biomonitoring studies indicate population-wide exposure, with a mean daily intake of 0.5 μg/kg body weight [12], and concentrations in processed foods can reach up to 4000 mg/kg [13]. In the absence of global regulatory limits, these findings underscore a substantial public health concern.

Despite extensive documentation of its neurotoxicity, the potential role of ACR in psychiatric disorders, particularly depression, remains underexplored. Major depressive disorder (MDD), characterized by persistent emotional dysregulation and functional impairment, affects 3.8% of the global population [14], with prevalence rising to 20% among adolescents [15]. Given MDD’s association with diminished psychosocial functioning and increased suicide risk [16], identifying preventable environmental risk factors like ACR exposure represents a public health priority. However, establishing a causal relationship between dietary ACR exposure and MDD has been challenging due to inherent limitations of observational studies, such as confounding and reverse causality. Moreover, the underlying molecular mechanisms linking ACR to depression have yet to be systematically explored.

While GA formation is well-established as a key mechanism for ACR’s carcinogenicity [17], its contribution to ACR-induced neurotoxicity and depression remains poorly defined. To bridge this knowledge gap, we hypothesized that dietary acrylamide (ACR) promotes depression through its epoxide metabolite GA, which mediates the suppression of SIRT3 function, leading to mitochondrial oxidative injury. To test this, we employed an integrative computational biology approach with the following aims: to investigate the potential causal effect of ACR on MDD risk using Mendelian randomization (MR); to identify the core protein targets and biological pathways through network toxicology and transcriptomic analysis; and to specifically test the novel hypothesis that SIRT3-mediated mitochondrial oxidative injury is a central mechanism linking ACR to depression, using molecular docking and dynamics simulations. Our work provides a foundational hypothesis-generating framework for future experimental validation and risk assessment. This comprehensive workflow (Figure 1) pioneers a novel approach to repositioning ACR-induced depression within the oxidative stress paradigm.

## 2. Materials and Methods

### 2.1. Target Construction and Toxicity of ACR

The 2D and 3D structures and SMILES notation of ACR were retrieved from the PubChem database (https://pubchem.ncbi.nlm.nih.gov/, accessed on 27 October 2024). Subsequently, ACR’s toxicological profile was predicted using in silico platforms including the ProTox 3.0 (https://tox.charite.de/, accessed on 27 October 2024) and ADMETlab 2.0 (https://admetmesh.scbdd.com/, accessed on 27 October 2024) 

### 2.2. Identification of Putative Protein Targets for ACR

The putative protein targets of ACR were identified by performing systematic queries against the ChEMBL, STITCH, and Swiss Target Prediction databases. First, in ChEMBL (https://www.ebi.ac.uk/chembl/, accessed on 27 October 2024), a compound search for “acrylamide” was performed with species filter set to Homo sapiens. Retrieved UniProt Accession IDs were then used to obtain experimentally confirmed gene nomenclature via UniProt (https://www.uniprot.org/, accessed on 27 October 2024). Next, the STITCH database (http://stitch.embl.de/, accessed on 27 October 2024) was queried using ACR’s SMILES (minimum interaction score: 0.4; species: Homo sapiens) to generate ACR–gene interaction networks. Finally, Swiss Target Prediction (http://swisstargetprediction.ch/, accessed on 27 October 2024) identified potential targets (probability > 0). After deduplication, a Venn diagram was constructed using the “ggvenn” (version 0.1.10) R package.

### 2.3. Collection of Depression-Related Genes

Depression-associated genes were retrieved from GeneCards (https://www.genecards.org/, accessed on 17 November 2024), OMIM (https://omim.org/, accessed on 17 November 2024), and Therapeutic Target Database (TTD, http://db.idrblab.net/ttd/, accessed on 17 November 2024) using “depression” as the query term. The inclusion criteria were: relevance score > 1.0 in GeneCards and comprehensive inclusion of all entries from OMIM and TTD. Following deduplication, a Venn diagram was generated in R software.

### 2.4. Identification of Shared Targets Between ACR and Depression

To identify the potential mechanistic links between ACR exposure and depression, the intersection between ACR-associated targets and depression-related genes was analyzed using R software through the “ggvenn” package, identifying shared ACR’s target genes and depression-associated genes, subsequently constructing edge lists and annotation matrices.

### 2.5. ACR-Depression Core Target Identification and PPI Network Construction

Protein interaction networks for shared targets were constructed via STRING (https://string-db.org/, accessed on 19 November 2024; confidence score > 0.15; species: Homo sapiens). Cytoscape v3.10.2 was used to visualize networks and subsequently identify hub proteins based on a combination of betweenness centrality and degree metrics.

### 2.6. GO and KEGG Pathway Enrichment Analyses

Functional annotation of 12 targets was conducted using R packages (“clusterProfiler”, “org.Hs.eg.db”). Significance was defined as *p* < 0.05 and FDR < 0.05. Top enriched terms were selected: the 10 most significant biological processes and 30 most enriched pathways, ranked by ascending FDR or gene count.

### 2.7. Molecular Docking

The three-dimensional structures of core proteins, including TP53 (PDB ID: 1AIE), PTGS2 (PDB ID: 5IKQ), JUN (PDB ID: 5FV8), and CASP3 (PDB ID: 1NME) were obtained from UniProt and Protein Date Bank (PDB) database (https://www.rcsb.org/, accessed on 22 November 2024). The ligands—ACR, its primary metabolite GA, and the negative control compound Dimethyl Sulfoxide (DMSO)—were sourced from the PubChem database. Receptor preparation was performed using PyMOL, involving the removal of solvent molecules and non-relevant ligands, followed by export in PDB format. Ligand preparation was conducted using AutoDock Tools 1.5.6, which included: addition of hydrogen atoms, identification of rotatable bonds, assignment of Gasteiger charges, and conversion to PDBQT format. Molecular docking was performed globally using AutoDock Vina, with a grid box sized to cover the entire protein structure for each target. For each protein–ligand pair (GA, ACR and DMSO), up to 20 conformational poses were generated and ranked based on binding affinity (kcal/mol). The pose with the most favorable binding energy and rational interaction geometry was selected for further analysis. All complexes were visualized and analyzed using PyMOL and Discovery Studio 2019.

### 2.8. Molecular Dynamics Simulation

Following molecular docking, the top-ranked complexes for each protein–ligand combination (GA, ACR and DMSO) underwent all-atom molecular dynamics (MD) simulations using AMBER22. Atomic charges for the ligands (GA, ACR and DMSO) were derived via HF/6-31G* level calculations performed with Gaussian 09, followed by RESP fitting for charge refinement. Parameterization utilized the GAFF2 force field for ligands and the ff14SB force field for proteins [18]. The LEaP module prepared the system by: adding hydrogens, solvating in TIP3P water (10 Å buffer), neutralizing with Na^+^/Cl^−^ ions, and generating topology files.

The simulation protocol comprised four stages: Firstly, energy minimization using 2500 steps of steepest descent followed by 2500 steps of conjugate gradient optimization; secondly, 200 ps NVT heating to 298.15 K; thirdly, two-phase equilibration—500 ps NVT (298.15 K) for solvent distribution and 500 ps NPT (1 atm) for density stabilization; finally, production MD under NPT conditions (298.15 K, 1 atm) with 2 fs timestep, employing PME for long-range electrostatics, SHAKE-constrained hydrogen bonds, Langevin temperature coupling (γ = 2 ps^−1^) [19], and 10 Å non-bonded cutoff. Trajectory analysis quantified protein–ligand interaction stability through Root mean square deviation (RMSD), Root mean square fluctuation (RMSF), radius of gyration (Rg), and hydrogen bond occupancy metrics.

### 2.9. Random Sample Gene Expression Validation

Gene expression matrices (GSE54567 and GSE54568) were obtained from the GEO database using the search query: “depression” AND “Homo sapiens”. Differential expression of core genes was assessed by two-tailed Student’s *t*-test (*p* < 0.05), with results visualized as box-and-whisker plots.

### 2.10. GWAS Summary Statistics of Depression and Core Protein

Depression GWAS summary statistics were obtained from IEU GWAS database (https://gwas.mrcieu.ac.uk/, accessed on 21 July 2025). Genetic instruments met genome-wide significance (*p* < 5.00 × 10^−8^) and independent of each other (linkage disequilibrium (LD) r^2^ < 0.001 within 10,000 kb, N = 26152). Depression GWAS data were sourced from the FinnGen consortium (https://r12.finngen.fi/, accessed on 21 July 2025, 149,403 cases and 111,976 controls). Instrument strength was assessed via F-statistics with SNPs of F < 10 excluded. Causal estimates were derived primarily using the inverse-variance weighted (IVW) method with a random-effects model, with MR-Egger and weighted median methods serving as supplementary analyses to test robustness, all implemented in the TwoSampleMR R package. Pleiotropy was evaluated using the MR-Egger intercept test, with heterogeneity quantified by Cochran’s Q statistic (*p* < 0.05 considered significant) and funnel plots. Leave-one-out analysis validated result consistency. MR-Pleiotropy Residual Sum and Outlier (MR-PRESSO) methods were also performed to evaluate the consistency of the conclusions.

### 2.11. Target-Specific Molecular Docking and Dynamics for SIRT3 and PTK2

Based on transcriptomic evidence identifying SIRT3 as differentially expressed and MR data establishing a causal association of PTK2 with depression, we performed molecular docking and dynamics simulations of SIRT3 and PTK2 with ligands (GA, ACR, and DMSO). The protocols followed identical procedures as described in Section 2.7 and Section 2.8, ensuring methodological consistency and enabling cross-study comparability. Specifically, the same preparation, docking, simulation parameters, and analysis methods were applied to these targets.

## 3. Results

### 3.1. In Silico Evaluation of ACR Toxicity and ADMET Properties

To corroborate the well-established toxicological profile of acrylamide (ACR) and to validate the predictive performance of our computational platforms, we employed the PROTox and ADMETlab tools. As consistently reported in decades of toxicological studies [20], ACR is a known neurotoxicant and carcinogen. Our in silico predictions aligned perfectly with these established findings: PROTox identified neurotoxicity and carcinogenicity as primary endpoints, with a predicted LD_50_ of 107 mg/kg (Toxicity Class III), which is consistent with previously reported rodent oral LD_50_ values ranging from 78.1–149.1 mg/kg of body weight [20]. Similarly, ADMETlab predictions confirmed significant potential for acute oral toxicity and carcinogenicity. This strong agreement validates the computational tools and provides a solid foundation for our subsequent investigation into the novel mechanism linking ACR to depression.

### 3.2. Network Toxicology Analysis of ACR Against Depression-Associated Targets

ACR-associated target genes were systematically identified by mining multiple bioinformatic databases (ChEMBL, STITCH, SwissTargetPrediction), yielding 34 unique molecular targets. Concurrently, disease-related targets for depression were extracted from GeneCards, OMIM, and TTD databases, retaining 5496 non-redundant targets after deduplication. Venn analysis revealed 12 overlapping targets, which were subsequently prioritized as shared candidate mechanisms linking ACR exposure to depression pathogenesis (Figure 2). The 12 overlapping targets were: *TP53*, *PTGS2*, *JUN*, *CASP3*, *ACE*, *GSR*, *SIRT3*, *SIRT2*, *CYP2E1*, *PTK2B*, *PTK2* and *MAP2*. Cytoscape 3.10.2 software was used to create an ACR-target-depression network diagram (Figure 3).

### 3.3. Screening of Key Targets and Construction and Analysis of the Protein–Protein Interaction

Subsequently, network interaction datasets and node attribute matrices were imported into Cytoscape for topological analysis. The protein–protein interaction network was visualized (Figure 4). Furthermore, four core regulators (*TP53*, *PTGS2*, *JUN*, *CASP3*) were prioritized using degree thresholding (degree ≥ 11). Notably, *SIRT3* and *PTK2*, despite their slightly lower topological significance (degree = 8 and 6, respectively), were retained for further investigation based on their well-established roles in mitochondrial dysfunction and synaptic plasticity, both of which are key pathways in depression pathogenesis.

### 3.4. GO and KEGG Pathway Enrichment Analysis

To elucidate the systemic mechanism underlying ACR’s neurotoxic effects on depression, Gene Ontology (GO) and Kyoto Encyclopedia of Genes and Genomes (KEGG) enrichment analyses were performed on 12 overlapping targets. The analysis identified 579 significant GO terms (false discovery rate (FDR) < 0.05), including 501 biological processes (BP), 17 cellular components (CC), and 61 molecular functions (MF). GO terms were analyzed using a dual-strategy approach: they were ranked based on both their false discovery rate (FDR) and gene ratio. The top 10 terms from each Gene Ontology category (BP, CC, MF) were selected for visualization. FDR-prioritized selection (Figure 5A,B) revealed BP predominantly associated with oxidative stress response (FDR = 1.6 × 10^−6^), CC localization in apical dendrites (FDR = 3.6 × 10^−3^), and MF involving NAD^+^ binding (FDR = 4.2 × 10^−3^). Gene ratio-weighted analysis (Figure 5C) demonstrated BP enrichment in oxidative stress regulation (Gene ratio = 0.58), CC clustering in neuronal cell bodies (Gene ratio = 0.33), and MF specialization in ubiquitin–protein ligase binding (Gene ratio = 0.25). Additionally, the top 30 signaling pathways were identified based on false discovery rate (FDR) and gene count, and visualized using histograms (Figure 5D) and lollipop charts (Figure 5E). In the neurotrophin signaling pathway (hsa04722), upregulation of p53 (*TP53*) and c-Jun (*JUN*) was observed (Figure 5F), while *Cox* (*PTGS2*) and *CASP3* were upregulated in the serotonergic synapse pathway (hsa04726) (Figure 5G).

### 3.5. Randomized Dataset Validation of Core Genes

The GSE54567 and GSE54568 datasets were randomly selected to validate expression profiles of core genes. These datasets comprise human dorsolateral prefrontal cortex samples from 29 MDD patients and 29 controls. Among the 12 shared targets, *TP53*, *JUN*, *CASP3*, *PTGS2* showed no significant expression changes (*p* > 0.05), whereas *SIRT3* exhibited marked dysregulation (*p* < 0.01) and *PTK2* demonstrated no significant difference (*p* > 0.05) in MDD cohorts (Figure 6).

### 3.6. Investigating the Causal Relationship Between ACR with Depression Using Mendelian Randomization

As shown in Table 1, MR analysis demonstrated significant causal effects of depression-related exposures on target genes: Inverse-variance weighted (IVW) model estimates for “Antidepressant”: *JUN*: OR = 1.029, 95% CI = 1.002–1.057, *p* = 0.034; *PTK2*: OR = 1.040, 95% CI = 1.005–1.076, *p* = 0.027; IVW estimates for “Depression–dysthymia” diagnosis: *JUN*: OR = 1.048, 95% CI = 1.021–1.076, *p* = 0.0004; *PTK2*: OR = 1.073, 95% CI = 1.039–1.109, *p* = 1.808 × 10^−5^. The scatter plots were shown in Figure 7. MR-Egger regression revealed no significant pleiotropy (*p* > 0.05 for all intercepts): “Antidepressant”: *JUN* intercept = 0.0034 (*p* = 0.622), *PTK2* intercept = −0.0025 (*p* = 0.736); “Depression–dysthymia”: JUN intercept = −0.009 (*p* = 0.123), *PTK2* intercept = 0.0013 (*p* = 0.836); Cochran’s Q-test indicated no heterogeneity (All *p* > 0.05): “Antidepressant”: *JUN* (*p* = 0.729), *PTK2* (*p* = 0.265); “Depression–dysthymia”: *JUN* (*p* = 0.722), *PTK2* (*p* = 0.484). Funnel plots exhibited symmetry (Figure 8), validating precision. Leave-one-out analysis and MR-PRESSO confirmed no influential outliers (Figure 9).

### 3.7. Molecular Docking of ACR, Its Metabolite GA, and Negative Control DMSO with Core Targets

We expanded our molecular docking simulations to include GA, the primary physiologically relevant metabolite of ACR, and DMSO as a negative control for non-specific weak binding. Simulations were performed against the core targets identified through our multi-omics integrative approach: TP53, JUN, CASP3, and PTGS2 (from network toxicology); SIRT3 (the sole significant differentially expressed protein from transcriptomics); and PTK2 (a Mendelian randomization-validated causal protein for depression).

The binding affinities for all protein–ligand complexes are summarized in Figure 10. Consistent with its role as a negative control, DMSO exhibited only weak, non-specific binding across all six targets, with energies ranging from −2.9 to −2.1 kcal/mol. Crucially, the docking results revealed a distinct preferential binding pattern between ACR and its metabolite, GA. Specifically, the binding affinity of GA was stronger than that of ACR for four targets: SIRT3 (GA: −4.7 vs. ACR: −4.0 kcal/mol), JUN (GA: −3.5 vs. ACR: −3.2 kcal/mol), TP53 (GA: −3.1 vs. ACR: −3.0 kcal/mol), and CASP3 (GA: −4.0 vs. ACR: −3.4 kcal/mol). Conversely, PTK2 (ACR: −3.0 vs. GA: −2.9 kcal/mol) and PTGS2 (ACR: −4.0 vs. GA: −3.9 kcal/mol) demonstrated stronger binding affinities for ACR itself. In summary, GA exhibited superior binding to SIRT3, JUN, TP53, and CASP3, whereas ACR bound more strongly to PTK2 and PTGS2. Crucially, the strongest GA binding affinity was observed for SIRT3, which aligns with its significant dysregulation in the human depression transcriptome, suggesting SIRT3 as a primary target of GA-mediated toxicity. The molecular docking diagrams are shown in Figure 11.

Detailed characterization of specific interactions revealed distinct hydrogen bonding patterns for ACR and its metabolite, GA, while the negative control DMSO showed no stable hydrogen bonds. Detailed hydrogen bond distances are listed below (all within the optimal 2.0–2.8 Å range for stable interactions): SIRT3-GA complex: bonds with Gly147 (2.5 Å), Thr150 (2.2Å), Arg345 (2.3/2.6 Å) and Arg365 (3.3 Å); SIRT3-ACR complex: bonds with Gly147 (2.5 Å), Thr150 (2.1 Å), and Arg345 (2.6 Å) and Arg365 (3.2 Å); SIRT3-DMSO complex: bonds with Thr320 (2.8 Å) and Ser321 (2.2/2.8 Å); JUN-GA complex: bonds with Gln12 (2.3/2.4 Å), Glu15 (2.1 Å), and Arg16 (2.3 Å); JUN-ACR complex: bonds with Gln12 (2.2/2.3 Å), Glu15 (2.1 Å), and Arg16 (2.3 Å); JUN-DMSO complex: bonds with Arg16 (2.2/2.2 Å); PTK2-GA complex: bond with Ala948 (1.9/2.2 Å), and Tyr953 (2.4 Å); PTK2-ACR complex: Single bond with Arg1045 (2.7 Å); PTK2-DMSO complex: bonds with Arg1045 (2.4/2.8 Å);TP53-GA complex: bond with Phe341 (2.8 Å); TP53-ACR and TP53-DMSO complex exhibited no detectable hydrogen bonds, relying exclusively on van der Waals forces for binding stability; CASP3-GA complex: bonds with Arg64 (1.9/2.5 Å), His121 (2.5 Å), Gln161 (2.3 Å), and Arg207 (2.3 Å); CASP3-ACR complex: bonds with Arg64 (2.0 Å), Gln161 (2.1 Å), Ser205 (2.6 Å) and Arg207 (2.2 Å); CASP3-DMSO complex: bonds with Arg64 (1.9 Å), and Arg207 (2.0/2.3 Å); PTGS2-GA complex formed bonds with His227 (2.6 Å), Val229 (2.5 Å), Asn538 (2.4 Å), and Val539 (2.6 Å); PTGS2-ACR complex formed bonds with Asp125 (2.2/2.4Å), Thr129 (2.8 Å), Arg150 (2.4 Å); PTGS2-DMSO complex formed bonds with Asp125 (2.1 Å) and Arg469 (2.7 Å). The hydrogen-bonding analysis revealed distinct interaction profiles that aligned with the binding affinity data. Notably, GA formed unique and additional hydrogen bonds with key residues in SIRT3 (Arg365), CASP3 (Arg64, His121), and PTGS2, which underpins its superior binding affinity over ACR for these targets. In contrast, the interactions mediated by DMSO were transient and non-specific, consistent with its weak binding energy as a negative control.

### 3.8. Molecular Dynamics Simulation of the Core Targets

To mechanistically explain the superior binding affinity of glycidamide (GA) observed in molecular docking, we conducted 100 ns molecular dynamics (MD) simulations to evaluate the dynamic stability and binding modes of ACR, its metabolite GA, and the negative control DMSO across all six target proteins.

RMSD of the protein backbone was calculated to assess global stability (Figure 12). Lower RMSD values indicate higher protein stability and reduced conformational fluctuation [21]. A consistent trend emerged: GA complexes exhibited the highest structural stability, achieving the lowest equilibrium RMSD values. This was particularly evident with SIRT3, JUN, and CASP3. Conversely, ACR complexes showed higher RMSD values with greater fluctuations, indicating less stable binding. As expected, DMSO showed the poorest stability, with high, fluctuating RMSD values consistent with its weak, non-specific binding. These results strongly suggest that the metabolite GA forms more stable and rigid complexes with the target proteins than its parent compound ACR, providing a molecular dynamics rationale for its previously observed superior binding affinity.

The local flexibility of the protein backbone upon ligand binding was evaluated by calculating the RMSF for each residue (Figure 13). A consistent and notable trend was observed across all six target proteins. The complexes formed with GA exhibited significantly suppressed fluctuations, particularly in regions encompassing the binding sites and flexible loop domains. For example, in SIRT3 and TP53, key residues lining the active site showed a pronounced decrease in flexibility when bound to GA compared to both ACR and the control DMSO. This indicates that GA binding effectively “locks” these functionally important regions into a more rigid conformation. These RMSF results complement the global stability data from the RMSD analysis, demonstrating that the superior stability of GA complexes is achieved through specific interactions that significantly reduce local backbone flexibility, likely contributing to tighter binding and prolonged residence time.

Rg was analyzed to assess the global structural compactness and conformational stability of the protein–ligand complexes over the 100 ns simulation time (Figure 14). The Rg value reflects the overall compactness of a structure, with more stable values indicating tighter binding [21]. GA complexes consistently maintained the most compact and stable global structure across all six target proteins. The Rg values for GA-bound states rapidly converged to a stable equilibrium. For instance, in complexes with SIRT3 and CASP3, the Rg for GA was lower than those observed for the ACR complexes. This significant reduction indicates that GA binding promotes a tighter, more condensed protein conformation. In contrast, the ACR complexes displayed intermediate Rg values with greater fluctuations, suggesting a less effective induction of structural compactness. As anticipated, DMSO control complexes exhibited the highest and most variable Rg profiles, characterized by an inability to reach a stable plateau. This is consistent with the weak, non-specific binding of DMSO, which fails to constrain the native protein structure effectively. These Rg results complement the local (RMSF) and global (RMSD) stability metrics, conclusively demonstrating that GA binding induces a more rigid, stable, and compact conformation in diverse protein targets.

The stability and specificity of ligand–protein interactions were further investigated by monitoring the number of hydrogen bonds formed over the simulation time course (Figure 15). A striking and consistent trend was observed across all six target proteins. GA complexes sustained a significantly greater number of stable hydrogen bonds compared to the ACR complexes. The GA curves not only plateaued at a higher value but also exhibited minimal fluctuations, indicating persistent and robust interactions. For instance, in complexes with CASP3 and PTGS2, the GA-bound form maintained an average of 4–5 hydrogen bonds. In contrast, the ACR complexes, limited by a simpler chemical structure, formed fewer hydrogen bonds (typically 3–4 on average) with higher volatility in the time-course data, consistent with a less stable interaction network. As expected, the negative control DMSO profiles consistently showed average of 1–3 hydrogen bonds, conclusively demonstrating the absence of specific, high-affinity interactions and validating its role as an appropriate control. These results definitively identify the enhanced hydrogen bonding capacity of GA as the primary molecular mechanism responsible for its ability to form more stable, rigid, and compact complexes, thereby explaining its superior binding affinity over ACR.

Our molecular dynamics simulations demonstrated that GA formed markedly more stable complexes with all target proteins than its precursor ACR, as evidenced by lower RMSD, RMSF, Rg values and more persistent hydrogen bonds. The superior binding affinity of GA can be attributed to its distinct chemical properties. Specifically, the highly reactive and strained epoxide ring of GA exhibits greater electrophilicity than the vinyl group of ACR [22], enabling the formation of more robust covalent adducts and a denser hydrogen-bond network [23]. These results provide a molecular rationale for GA, rather than ACR, acting as the primary toxic effector mediating sustained protein inhibition and mitochondrial dysfunction [24].

## 4. Discussion

This integrated multi-omics and Mendelian randomization (MR) study provides the first population genetic evidence establishing dietary acrylamide (ACR) exposure as a risk factor for depression, primarily mediated through a SIRT3-dependent mitochondrial oxi-dative stress pathway. This finding extends the initial mechanistic hypothesis proposed by Li et al., which was demonstrated in hepatic models to the human brain, and provides genetic causal support for its role in neuropsychiatric pathophysiology [11]. Our computational approach delineated a coherent pathogenic cascade: network toxicology prioritized key targets; human prefrontal cortex transcriptomics specifically pinpointed SIRT3 as the most significantly suppressed gene in depression; and Mendelian randomization confirmed the genetic causality of downstream signaling orchestrators JUN and PTK2. Crucially, molecular dynamics simulations revealed that the neurotoxicity of ACR is likely executed by its highly reactive metabolite, GA, which forms a more stable complex with the central mediator SIRT3 than ACR itself, thereby providing a molecular rationale for this pathway.

The mechanistic pathway delineated by our multi-omics analyses—initiated by SIRT3 suppression and culminating in neuronal damage—finds compelling support in emerging epidemiological literature that translates preclinical acrylamide toxicity into human psychiatric outcomes. A recent nationwide population-based study demonstrated a linear positive association between urinary biomarkers of acrylamide metabolites (AAMA, GAMA) and depressive symptoms, specifically highlighting inflammatory response and oxidative stress as potential key mechanisms [25]. This observation at the population level directly corroborates the core pathological processes identified in our model. Furthermore, supporting evidence links frequent consumption of fried food, a major dietary source of acrylamide, to a higher prevalence of anxiety and depressive symptoms, purportedly through dysregulated lipid metabolism and inflammatory pathways [26]. While these epidemiological studies robustly link ACR exposure to clinical symptoms, they are inherently limited in establishing causality and elucidating underlying molecular mechanisms. Our integrated approach directly addresses this gap: we not only provide genetic causal evidence for the association using Mendelian randomization but also pinpoint SIRT3 inhibition as the critical molecular initiator and JUN/PTK2 as the causal mediators translating this mitochondrial damage into depressive pathogenesis. Thus, our work moves beyond correlation to propose a testable molecular and genetic pathway that explains how dietary acrylamide exposure could ultimately manifest as depression.

Our multi-omics approach, however, revealed a critical and informative discrepancy: while network toxicology identified *TP53*, *CASP3*, *PTGS2*, and *JUN* as top core targets, subsequent analyses refined this picture. Only *SIRT3* showed significant differential expression in the human depression transcriptome, and MR confirmed *JUN* and *PTK2*—but not *TP53*, *CASP3*, or *PTGS2*—as causal genetic risk factors. The genetic causality of JUN and PTK2 suggests that their activation in depression, upon ACR exposure, is likely an indirect consequence, mediated by the primary event of SIRT3 suppression and the resulting oxidative stress.

This informative discrepancy across omics layers does not invalidate the predicted roles of TP53, CASP3, and PTGS2; rather, it highlights the complexity of environmental toxicant actions. We propose three non-mutually exclusive explanations:

First, post-transcriptional dominance: For key apoptotic and inflammatory executors like PTGS2, TP53, and CASP3, their activity is predominantly regulated at the post-translational level (e.g., phosphorylation, cleavage, ubiquitination) rather than through mRNA abundance [22]. This premise is strongly supported by our molecular dynamics simulations, which demonstrated that GA forms highly stable complexes with these targets, suggesting a potential for direct functional modulation that bypasses transcriptional regulation.

Second, spatiotemporal specificity: The GEO transcriptomic datasets represent a snapshot of a chronic, multifactorial disease state, which may not capture the acute or sub-chronic molecular perturbations. This limitation is critical, as there can be a substantial difference in the time course of transcriptomic changes [27].

Finally, hierarchical vulnerability in the pathogenic cascade: Our data suggest a mechanistic hierarchy where SIRT3 suppression may be the primary, upstream event, leading to oxidative stress, which then secondarily activates downstream executors [24]. This could explain why genetic variants in TP53, CASP3, or PTGS2 were not significant in the MR analysis, as they are not the initiating factors.

Therefore, the multi-omics heterogeneity itself becomes a key finding, delineating a pathogenic cascade in which ACR exposure primarily impairs the mitochondrial gatekeeper SIRT3, with downstream consequences involving JUN/PTK2 signaling and the functional activation of apoptotic and inflammatory executors. Beginning with this primary mitochondrial target, we delve into the mechanistic roles of each component.

### 4.1. Mitochondrial Gatekeeper: SIRT3 as the Primary Metabolic Redox Target of ACR

As a mitochondrial NAD^+^-dependent deacetylase highly expressed in the brain, SIRT3 functions as a crucial metabolic stress sensor that couples cellular redox status to mitochondrial function [28]. We propose that this key protein represents the primary molecular target through which dietary acrylamide (ACR) disrupts neuronal oxidative homeostasis. The mechanism by which ACR likely impairs mitochondrial function is through the inhibition of SIRT3 activity, which critically regulates mitochondrial reactive oxygen species (mROS) homeostasis. SIRT3 achieves this primarily by activating superoxide dismutase 2 (SOD2), the major mitochondrial superoxide scavenger, via deacetylation of its lysine 68 residue, thereby enhancing its catalytic activity and promoting cell survival [29]. The significance of SIRT3 dysregulation in depression pathogenesis is underscored by its role in preserving neurogenesis [30]. Furthermore, SIRT3 activation alleviates depressive-like behaviors in rodent models by boosting mitochondrial energy metabolism and reducing ROS [31]. Supporting the therapeutic relevance of this pathway, SIRT3 activation has been shown to rescue mitochondrial dysfunction [32] and kaempferol exerts antidepressant effects via SIRT3 upregulation [33]. Our findings suggest that ACR primarily impairs this critical mitochondrial gateway by targeting SIRT3 function, initiating the downstream pathological cascade.

### 4.2. Signaling Orchestrators: JUN/PTK2 as Genetic Risk Factors Potentially Activated by Oxidative Stress

Our Mendelian randomization analysis provides the first genetic evidence that JUN and PTK2 are causal risk factors for depression. Although our study does not directly prove that ACR exposure activates these genes, their well-characterized roles as key sensors of cellular stress pathways position them as plausible downstream effectors. This is highly consistent with our model, as both JUN and PTK2 are known to be activated by oxidative stress [34,35] the primary consequence of ACR-induced SIRT3 inhibition. The JUN gene encodes c-Jun, a master regulator of neuronal death and regeneration [36]. Similarly, FAK (PTK2) activity is modulated by redox signaling [37]. In summary, the genetic causality established by MR, combined with the documented sensitivity of JUN/PTK2 pathways to oxidative stress, supports their inclusion in our model as key mediators linking ACR-induced mitochondrial injury to depressive pathogenesis.

### 4.3. Apoptotic Network: TP53/CASP3 as Potential Effectors of Oxidative Neuronal Injury

In addition to the SIRT3-JUN/PTK2 axis, our network toxicology approach identified TP53 and CASP3 as central nodes in ACR-induced neurotoxicity. It is important to emphasize that their predicted involvement is based on protein interaction networks, and their lack of transcriptional dysregulation in human depression datasets suggests a primary role at the post-translational level. This is consistent with their known mechanisms of action: TP53’s pro-apoptotic activity is often triggered by post-translational stabilization in response to DNA damage [38], while CASP3, the key executioner protease, is activated by proteolytic cleavage of its inactive zymogen [39]. Therefore, ACR-induced oxidative stress may lead to their functional activation without necessarily altering their mRNA abundance or attenuating oxidative neuronal death [40]. This notion is supported by evidence linking their activity to depression-related oxidative injury. For instance, dysregulation of the TP53 network may contribute to the pathogenesis of major depressive disorder [38]. Critically, oxidative stress severely affects the CNS due to high metabolic activity and limited regeneration. Elevated ROS levels induce oxidative macromolecular damage, which synergistically activates apoptosis and pro-inflammatory cascades [41]. Within this context, TP53 and CASP3 are hypothesized to act as critical downstream executors of apoptosis in our model, likely activated by the oxidative damage resulting from SIRT3 suppression.

### 4.4. Neuroinflammatory Hub: PTGS2 as a Downstream Mediator of Oxidative Stress

PTGS2 (prostaglandin-endoperoxide synthase 2), a key inducible enzyme in prostaglandin biosynthesis, functions as a critical proinflammatory mediator and is implicated in cytokine-induced depression. Notably, PTGS2-derived prostaglandin E2 (PGE2) in the hippocampal CA1 region can activate glial cells, establishing a proinflammatory feedback loop relevant to depressive pathophysiologys [42]. We hypothesize that PTGS2 induction may represent a pivotal downstream consequence of ACR-induced SIRT3 suppression and subsequent oxidative stress. This is mechanistically plausible given that oxidative stress is a well-established potent inducer of PTGS2 expression. The consequent sustained production of prostaglandins (e.g., PGE2) driven by PTGS2 is known to exacerbate neuroinflammatory processes, thereby potentially contributing to the manifestation of depressive-like behaviors [43]. Thus, PTGS2 is positioned within our model as a key molecular nexus linking oxidative damage to neuroinflammation. Its potential as a therapeutic target is further supported by interventions such as quercetin, which has been shown to ameliorate depression-related symptoms in breast cancer models through multi-faceted mechanisms that include the suppression of PTGS2 [44]. The findings presented in Figure 16 indicate that ACR induces depression via these molecular mechanisms.

Our integrated multi-omics analysis suggests a potential mechanistic link between dietary ACR exposure and the pathogenesis of depression, possibly mediated through SIRT3-dependent mitochondrial dysfunction, thereby addressing a significant knowledge gap in environmental neurotoxicology. This study, however, has several limitations. First, target prioritization relied on database-derived predictions, which are inherently susceptible to algorithmic bias. Second, further extensive in vivo validation is necessary to confirm the effects of chronic, low-dose ACR exposure that accurately mirrors typical human dietary intake. Finally, the generalizability of our findings is constrained by the exclusive use of European-ancestry cohorts, underscoring the need for validation in more diverse populations.

Collectively, our findings provide initial evidence that may help reconceptualize ACR-associated depression as an environmentally triggered disorder driven by oxidative stress, offering a hypothetical rationale for targeting SIRT3 to mitigate dietary neurotoxicity. These findings could guide future scientific discourse on food safety regulations and underscore the need for large-scale exposure assessment in high-consumption populations as further evidence emerges.

## 5. Conclusions

This multi-omics investigation establishes dietary acrylamide (ACR), a pervasive environmental toxicant in thermally processed foods, as a modifiable risk factor for depression. We elucidate that its toxicity is primarily executed by the highly reactive metabolite, glycidamide (GA), which directly suppresses SIRT3-mediated SOD2 deacetylation, triggering mitochondrial oxidative injury and activating causal pathways via JUN and PTK2, thereby converging TP53/CASP3-mediated apoptotic and PTGS2-driven neuroinflammatory cascades. These findings hold direct public health relevance, advocating for the monitoring of both the parent compound ACR and its metabolite GA in exposure assessments. Furthermore, they support biomarker-guided monitoring of JUN and PTK2 in high-risk processed food consumers and position SIRT3 activation as a promising therapeutic countermeasure against environmental neurotoxicity. To mitigate this preventable health burden, urgent priorities include validating low-dose ACR/GA exposure thresholds through in vivo studies, expanding multi-ethnic Mendelian randomization cohorts to ensure broader applicability, and implementing practical, large-scale surveillance programs to effectively monitor and reduce population-level exposure.

Furthermore, we acknowledge an important mechanistic limitation of our current study: while our molecular dynamics simulations provide strong computational evidence for the GA-SIRT3 interaction, direct biochemical validation of acrylamide-induced SIRT3 functional suppression is still lacking. To address this gap, we have designed a series of follow-up experiments including (1) in vitro exposure of neuronal cell lines (SH-SY5Y and primary cortical neurons) to physiologically relevant concentrations of ACR and GA, measuring SIRT3 deacetylase activity using fluorometric assays and monitoring mitochondrial acetylation status via Western blot; (2) in vivo validation using a rodent model of chronic low-dose ACR exposure that mimics human dietary intake patterns, with subsequent assessment of SIRT3 activity and mitochondrial function in brain tissue; and (3) structural validation through co-crystallization studies of GA bound to SIRT3. These planned investigations will provide crucial experimental confirmation of the proposed mechanism and strengthen the translational relevance of our findings.

## Figures and Tables

**Figure 1 cimb-47-00836-f001:**
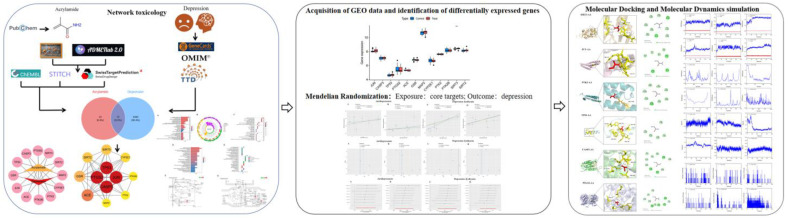
Flowchart of this study.

**Figure 2 cimb-47-00836-f002:**
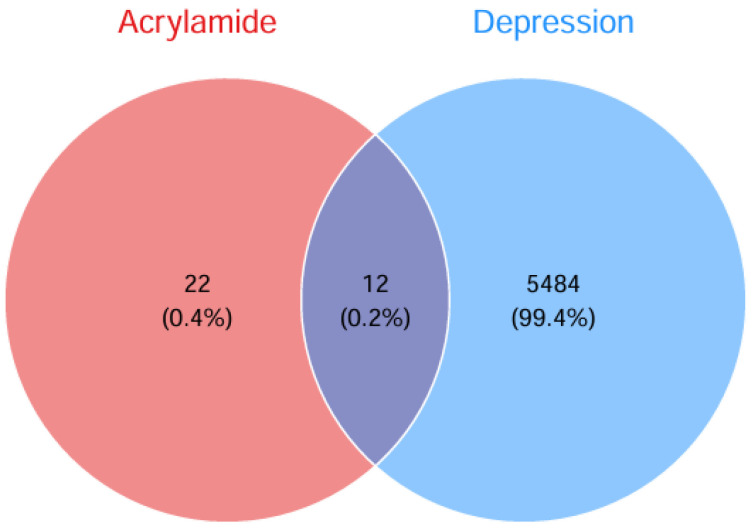
Venn diagram showing intersecting targets between ACR and depression.

**Figure 3 cimb-47-00836-f003:**
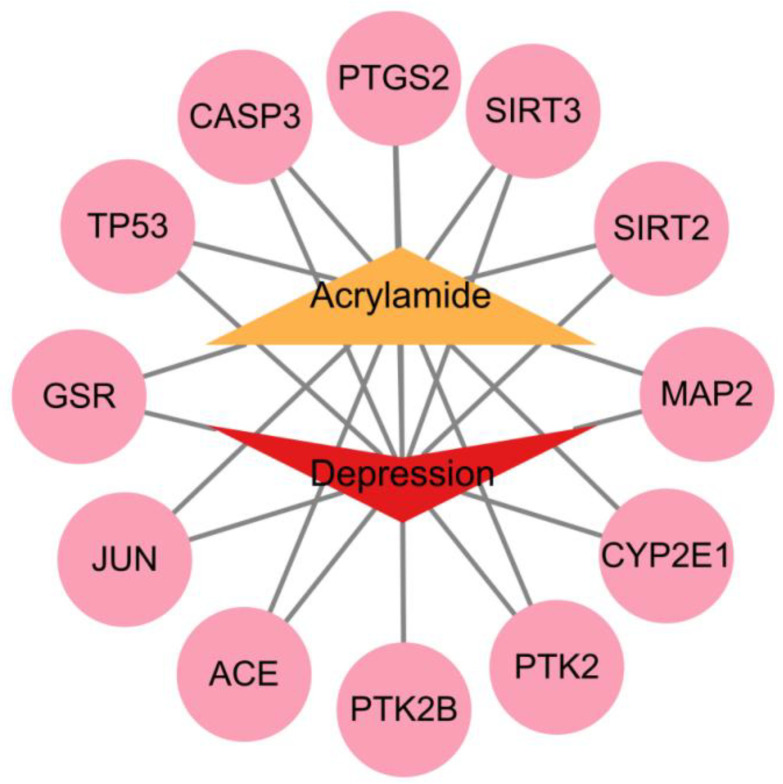
ACR-target-depression network diagram.

**Figure 4 cimb-47-00836-f004:**
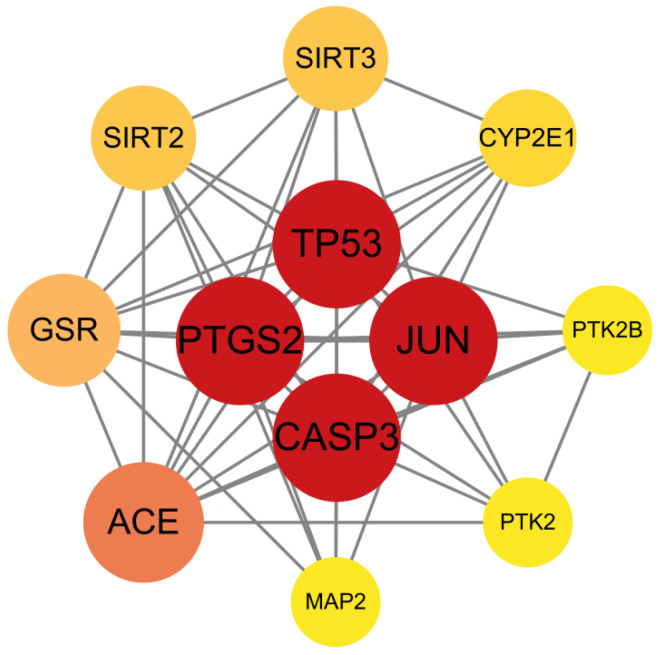
Protein–protein interaction (PPI) network of core targets. *TP53*, *JUN*, *CASP3*, and *PTGS2* were identified as the most highly connected hub nodes, as indicated by their red-colored vertices in the network diagram.

**Figure 5 cimb-47-00836-f005:**
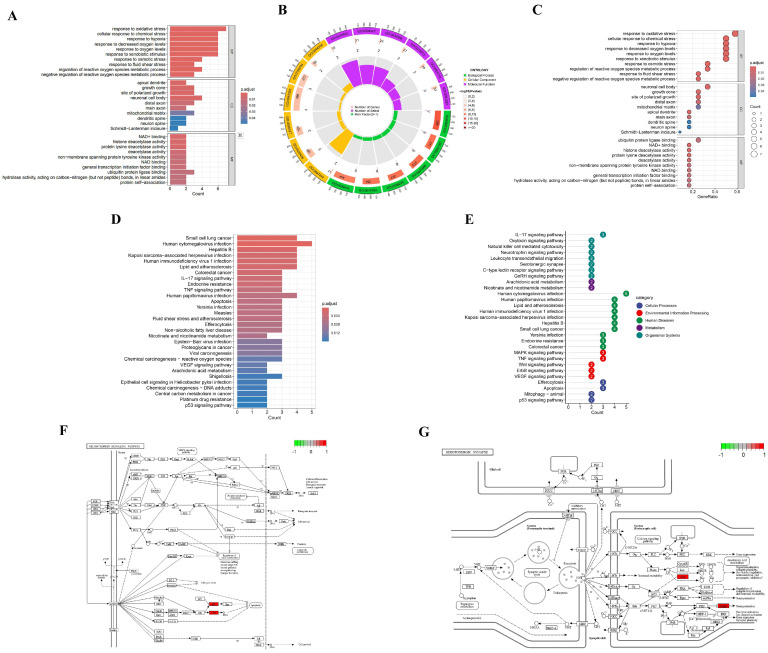
Functional enrichment analysis of GO and KEGG pathways. (**A**) GO histogram displaying the top 10 significantly enriched terms across BP, CC, and MF. Bar length corresponds to gene count; color intensity indicates statistical significance (red: Padjust < 0.01; blue: Padjust < 0.04). (**B**) Circular enrichment plot: outer ring labels enriched terms; inner ring quantifies gene numbers per term. (**C**) Bubble chart mapping pathway associations: bubble size represents gene quantity; color saturation denotes enrichment significance (red: Padjust < 0.01; blue: Padjust < 0.04). (**D**) KEGG enrichment histogram of the top 30 pathways. Bar length reflects gene enrichment frequency; color gradient encodes significance level (red: Padjust < 0.04; blue: Padjust < 0.012). (**E**) Lollipop plot visualizing pathway enrichment: vertical axis lists KEGG terms; horizontal position indicates enriched gene count. (**F**,**G**) Core target interactions within (**F**) neurotrophin and (**G**) serotonergic synapse signaling pathways. Red rectangles denote upregulated genes.

**Figure 6 cimb-47-00836-f006:**
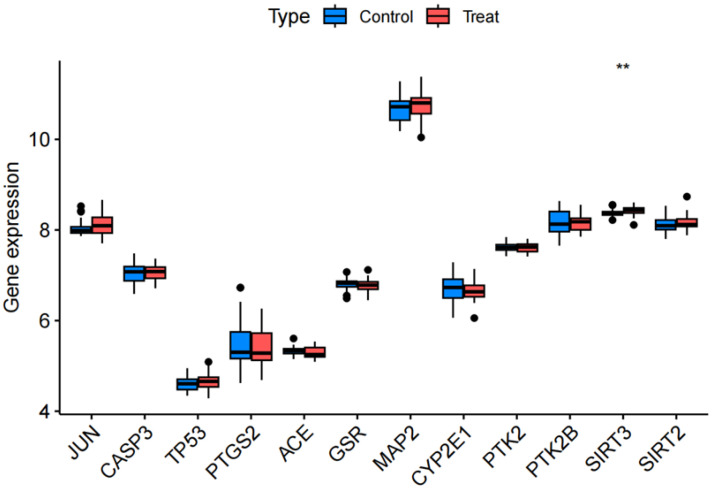
Differential expression profiles of 12 core genes across random data sets.** *p* < 0.01.

**Figure 7 cimb-47-00836-f007:**
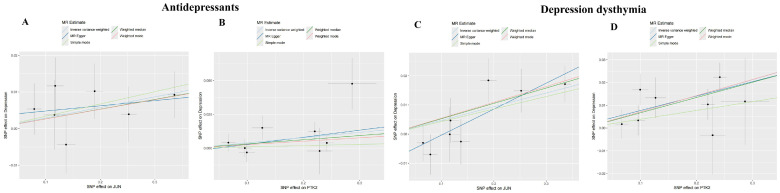
The scatter plots of MR analyses. (**A**) Analysis for “*JUN*” on “Antidepressant”. (**B**) Analysis for “*PTGS2*” on “Antidepressant”. (**C**) Analysis for “*JUN*” on “Depression–dysthymia”. (**D**) Analysis for “*PTGS2*” on “Depression–dysthymia”.

**Figure 8 cimb-47-00836-f008:**
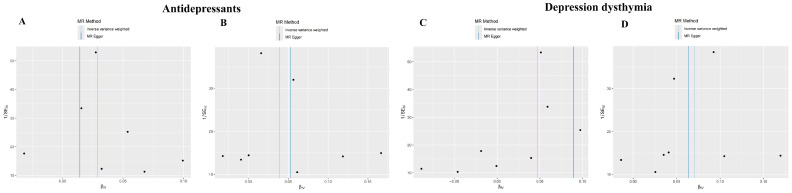
The funnel plots for assessing heterogeneity. (**A**) Analysis for “*JUN*” on “Antidepressant”. (**B**) Analysis for “*PTGS2*” on “Antidepressant”. (**C**) Analysis for “*JUN*” on “Depression–dysthymia”. (**D**) Analysis for “*PTGS2*” on “Depression–dysthymia”.

**Figure 9 cimb-47-00836-f009:**
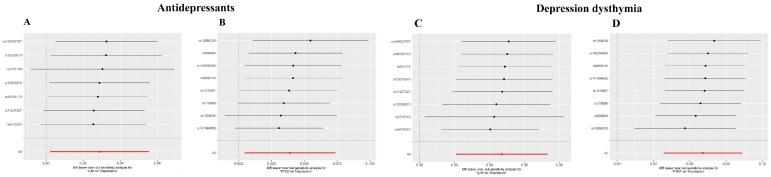
The leave-one-out plots for assessing the robustness. The horizontal red line represents the overall causal estimate using all instrumental variables (IVs). Each point depicts the causal estimate when the corresponding single nucleotide polymorphism (SNP) was omitted from the analysis. (**A**) Analysis for “*JUN*” on “Antidepressant”. (**B**) Analysis for “*PTGS2*” on “Antidepressant”. (**C**) Analysis for “*JUN*” on “Depression–dysthymia”. (**D**) Analysis for “*PTGS2*” on “Depression–dysthymia”.

**Figure 10 cimb-47-00836-f010:**
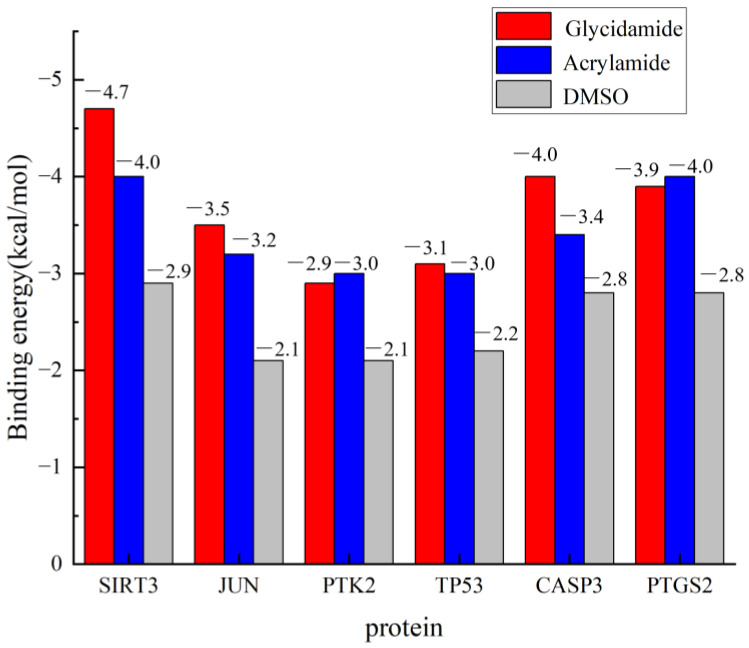
Comparative molecular docking binding affinities of ACR, GA, and DMSO with core target proteins. Binding energies (kcal/mol) for acrylamide (ACR, blue bars), its metabolite glycidamide (GA, red bars), and the negative control dimethyl sulfoxide (DMSO, gray bars) are shown.

**Figure 11 cimb-47-00836-f011:**
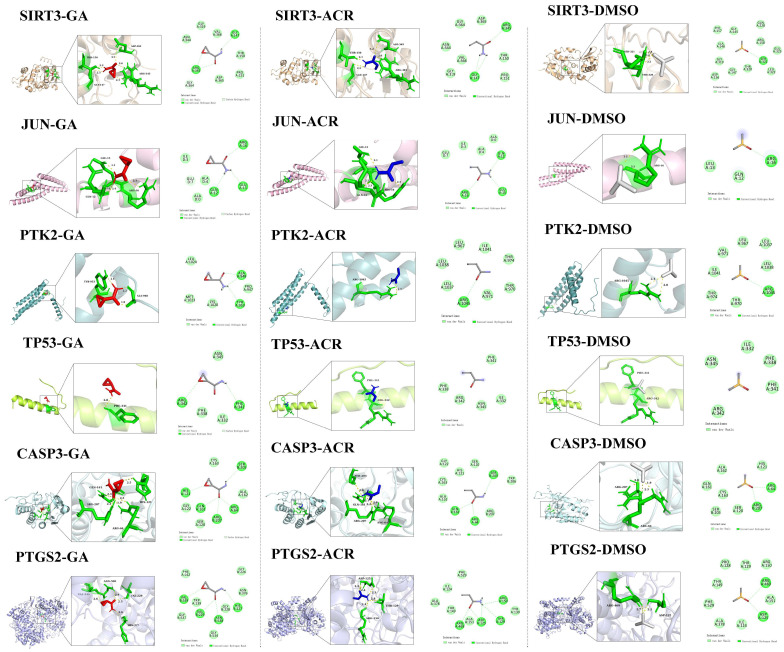
Molecular docking results of ACR, GA, and DMSO with SIRT3, JUN, PTK2, TP53, CASP3 and PTGS2 displaying the lowest binding energies in both 3D and 2D formats.

**Figure 12 cimb-47-00836-f012:**
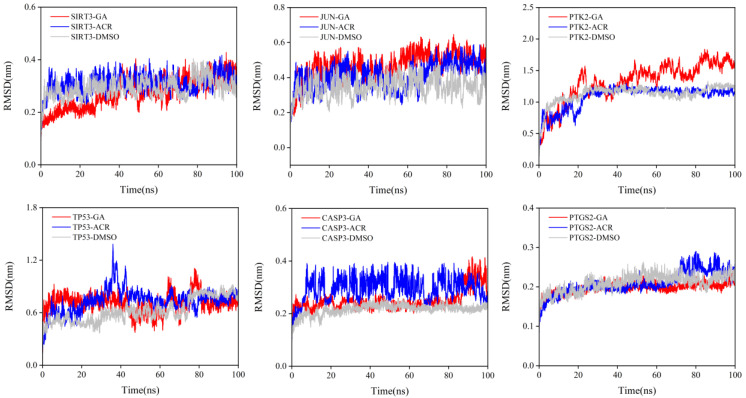
RMSD of the protein backbone during the 100 ns molecular dynamics simulations. The simulations were performed for the complexes of SIRT3, JUN, PTK2, TP53, CASP3, and PTGS2 with GA (red), ACR (blue), and DMSO (gray).

**Figure 13 cimb-47-00836-f013:**
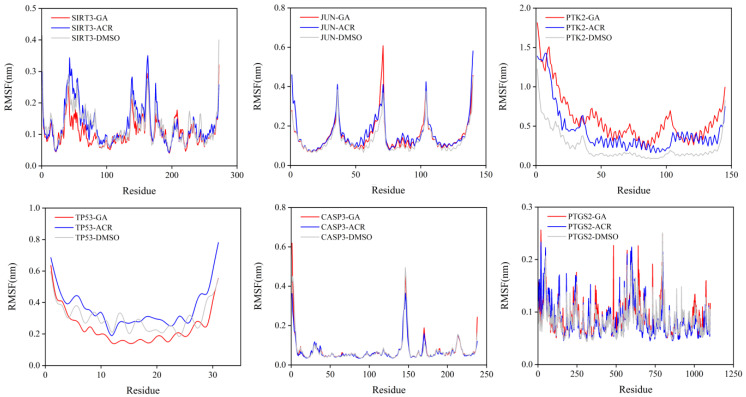
RMSF of the protein backbone during the 100 ns molecular dynamics simulations. The simulations were performed for the complexes of SIRT3, JUN, PTK2, TP53, CASP3, and PTGS2 with GA (red), ACR (blue), and DMSO (gray).

**Figure 14 cimb-47-00836-f014:**
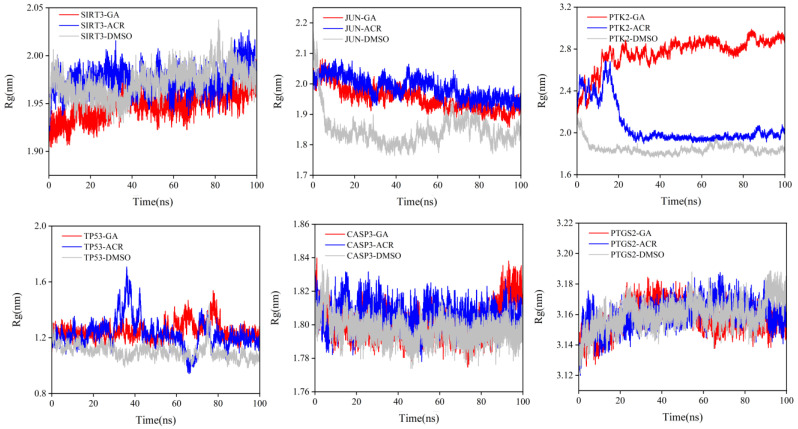
Rg trajectories for SIRT3, JUN, PTK2, TP53, CASP3 and PTGS2 complexed with GA (red), ACR (blue), and DMSO (gray) are shown.

**Figure 15 cimb-47-00836-f015:**
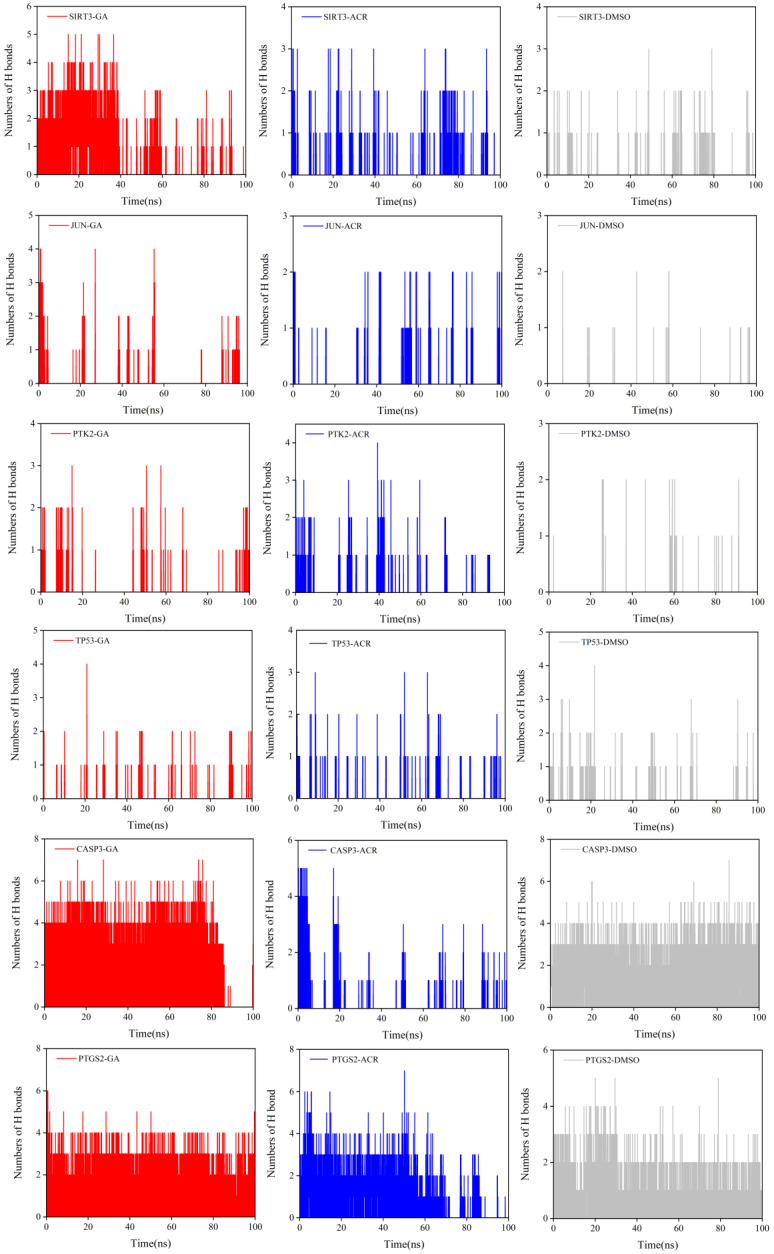
The number of hydrogen bonds for complexes of SIRT3, JUN, PTK2, TP53, CASP3, and PTGS2 with GA (red), ACR (blue), and DMSO (gray) are shown.

**Figure 16 cimb-47-00836-f016:**
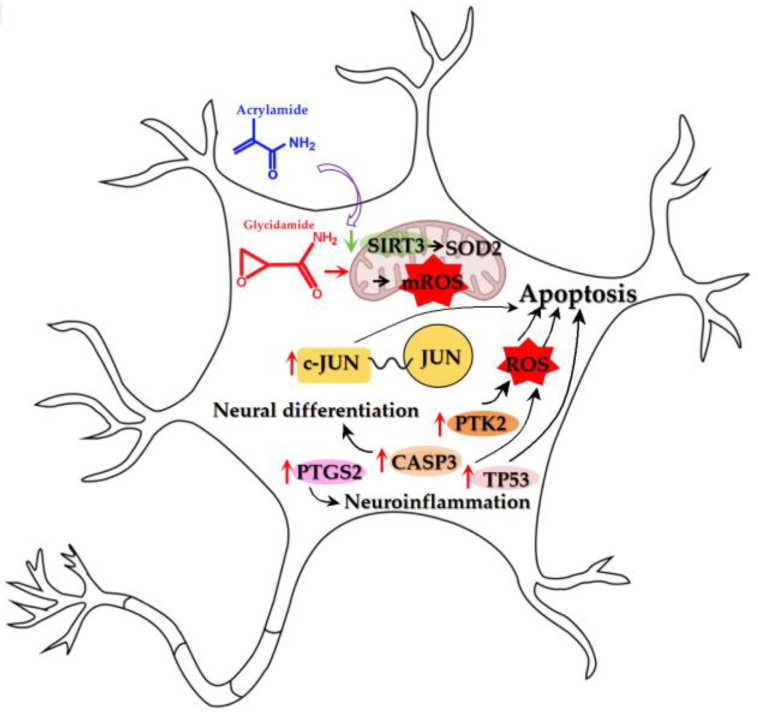
Schematic diagram showing molecular mechanisms of ACR-induced depression. A curved arrow indicates induction or causation. A green arrow signifies downregulated expression. A red arrow signifies upregulated expression.

**Table 1 cimb-47-00836-t001:** The results of each MR estimates.

MR Analyses	*JUN* OR (95% CI)	*p*	*PTK2* OR (95% CI)	*p*
Antidepressants				
MR Egger	1.014 (0.955–1.078)	0.664	1.054 (0.969–1.147)	0.268
Weighted median	1.027 (0.995–1.060)	0.095	1.031 (0.990–1.075)	0.140
Inverse variance weighted	1.029 (1.002–1.057)	0.034	1.040 (1.005–1.076)	0.027
Simple mode	1.034 (0.986–1.085)	0.215	1.009 (0.943–1.080)	0.795
Weighted mode	1.027 (0.994–1.061)	0.163	1.026 (0.980–1.074)	0.313
Depression–dysthymia				
MR Egger	1.093 (1.037–1.153)	0.016	1.065 (0.989–1.148)	0.145
Weighted median	1.054 (1.021–1.088)	0.001	1.069 (1.026–1.115)	0.0017
Inverse variance weighted	1.048 (1.021–1.076)	0.0004	1.073 (1.039–1.109)	1.808 × 10^−5^
Simple mode	1.044 (0.994–1.096)	0.129	1.039 (0.969–1.115)	0.315
Weighted mode	1.056 (1.020–1.093)	0.018	1.074 (1.024–1.125)	0.021

## Data Availability

The raw data supporting the conclusions of this article will be made available by the authors on request.

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
