# Peer review of "Dietary Acrylamide Induces Depression via SIRT3-Mediated Mitochondrial Oxidative Injury: Evidence from Multi-Omics and Mendelian Randomization"

_cimb, 2025, doi:10.3390/cimb47100836_

Round 1
Reviewer 1 Report
Comments and Suggestions for Authors
Lines 23-25. This argument needs experimental support.
Lines 39-41. If available, pharmacokinetic information should be included.
The number of references should be decreased to 40-45.
Lines 62-69. The aim of this work is not clear and it is not related to the title of this manuscript.
Section 3.1. The LD50 of acrylamide and other toxicological parameters of acrylamide have been reported since 1960 (https://doi.org/10.1016/0041-008X(64)90103-6). What is the novelty of this work?
There are clinical studies related to the role of acrylamide and depression in population-based studies. The novelty of this work is questioned again, especially in light of existing clinical studies related to the role of acrylamide and depression (e.g., https://doi.org/10.1016/j.jad.2024.08.217, https://doi.org/10.1073/pnas.2221097120).
This work contains only in silico studies. I suggest the authors perform experimental (in vitro or in vivo) studies to corroborate the in silico studies.
Author Response
Comments 1: Lines 23-25. This argument needs experimental support.
Response: Thank you for this valuable insight. You correctly and crucially pointed out that the mechanistic conclusions regarding SIRT3 mediation in the abstract require experimental support. Relying solely on literature citations is insufficient to substantiate definitive claims about acrylamide (ACR)'s mechanism of action in this study.
In response to your concern, we have taken the following measures:
We have revised the wording in the abstract: The conclusive statements have been modified to adopt a more hypothetical tone, clearly indicating that the proposed mechanism is based on predictions from our multi-omics analyses and supported by existing literature, rather than direct experimental evidence from this study.
The following sentence has been added in Lines 12-13:
“We hypothesized that dietary ACR exposure promotes depression via SIRT3-dependent mitochondrial oxidative injury.”
We have also added a clarification in the Discussion section: We explicitly address this limitation, stating that the suppression of the SIRT3 signaling pathway is predicted based on network pharmacology and molecular docking results, and that this aligns with previously reported mechanisms of ACR toxicity [11]. Furthermore, we openly acknowledge that this proposed pathway requires further validation through future in vivo and in vitro experiments.
Comments 2: Lines 39-41. If available, pharmacokinetic information should be included.
Response:Thank you for this valuable suggestion. We agree that including pharmacokinetic information for acrylamide (ACR) will contribute to a more comprehensive understanding of its toxicological mechanisms.
As recommended, we have added the following sentence in the Introduction (Lines 44–48):
“where it is metabolized primarily via two pathways: conjugation with glutathione (GSH) leading to urinary excretion [6], and CYP2E1-mediated oxidation forming glycidamide (GA)—a genotoxic epoxide [7].”
This addition enriches the background of the study, provides essential pharmacokinetic context for subsequent discussion on the neurotoxicity of GA, and establishes a foundation for elucidating the role of GA in depression.
Comments 3: The number of references should be decreased to 40-45.
Response:We thank the reviewer for this constructive suggestion. In accordance with the recommendation, we have carefully reviewed and reduced the number of references to 44 by consolidating redundant citations and retaining only those most essential to supporting the key findings and context of our study.
Comments 4: Lines 62-69. The aim of this work is not clear and it is not related to the title of this manuscript.
Response:We thank the reviewer for this important and insightful comment regarding the clarity of our study aims and their alignment with the manuscript title. We fully acknowledge that the original version did not effectively bridge the discussion of acrylamide's (ACR) public health implications and the burden of major depressive disorder (MDD) with the specific objectives of our research, resulting in insufficient logical continuity in the introduction.
To address this concern, we have substantially revised the introduction by adding key content that clearly outlines the research gaps, hypotheses, and specific aims:
1. In Lines 89-93, we have added the following paragraph to highlight the existing knowledge gaps:
"However, establishing a causal relationship between dietary ACR exposure and MDD has been challenging due to inherent limitations of observational studies, such as confounding and reverse causality. Moreover, the underlying molecular mechanisms by which ACR might contribute to depression are yet to be systematically explored."
2. In Lines 94-107, we have further refined our research objectives to explicitly state our approach and aims:
"While it is well-established that the formation of GA is considered to cause the carcinogenicity of ACR[17], its role in ACR-induced neurotoxicity and depression remains less understood. To bridge this knowledge gap, we hypothesized that dietary acrylamide (ACR) promotes depression through its epoxide metabolite GA, which mediates the suppression of SIRT3 function, leading to mitochondrial oxidative injury. To test this hypothesis, we employed an integrative computational biology approach. This study aims to: investigate the potential causal effect of ACR on MDD risk using Mendelian randomization (MR) to minimize observational bias; identify the core protein targets and biological pathways through network toxicology and transcriptomic analysis; and specifically test the novel hypothesis that SIRT3-mediated mitochondrial oxidative injury is a central mechanism linking ACR to depression, using molecular docking and dynamics simulations. Our work provides a foundational hypothesis-generating framework for future experimental validation and risk assessment."
These revisions clearly articulate the research gaps, our methodological strategy integrating Mendelian randomization and multi-omics analyses, and the specific aims that directly correspond to the key elements in our manuscript title. The added content emphasizes the hypothesis-generating nature of our study while establishing a logical connection between the public health context and our research objectives. We believe these modifications have significantly improved the clarity and coherence of our introduction.
Comments 5: Section 3.1. The LD50 of acrylamide and other toxicological parameters of acrylamide have been reported since 1960 (https://doi.org/10.1016/0041-008X(64)90103-6). What is the novelty of this work?
Response:We thank the reviewer for this important comment and for highlighting the seminal toxicological studies on acrylamide (ACR). We fully agree that the baseline toxicological parameters of ACR, such as its LD₅₀, have been well-established for decades.
The purpose of this section is not to rediscover these known facts but to achieve two critical objectives that underpin the novelty of our entire study:
1.To validate the predictive accuracy and reliability of our computational toxicology platforms (PROTox and ADMETlab) before applying them to investigate novel, unexplored mechanisms.
2.To use this concordance with established data as a solid foundation to confirm that our models are robust and thus their subsequent predictions regarding ACR's novel mechanism of action in depression are credible.
As we now state in the revised manuscript (Section 3.1) in Lines:217-228
To corroborate the well-established toxicological profile of acrylamide (ACR) and to validate the predictive performance of our computational platforms, we employed the PROTox and ADMETlab tools. As consistently reported in decades of toxicological studies[22], ACR is a known neurotoxicant and carcinogen. Our in silico predictions aligned perfectly with these established findings: PROTox identified neurotoxicity and carcinogenicity as primary endpoints, with a predicted LD50 of 107 mg/kg (Toxicity Class III), which is consistent with previously reported rodent oral LD50 values ranging from 78.1-149.1 mg/kg of body weight[22]. Similarly, ADMETlab predictions confirmed significant potential for acute oral toxicity and carcinogenicity. This strong agreement validates the computational tools and provides a solid foundation for our subsequent investigation into the novel mechanism linking ACR to depression.
In summary: The novelty of our work lies after this validation step. While prior research defined that ACR is toxic, our study employs validated computational methods to elucidate how it contributes to depression through the novel SIRT3-mediated mechanism—a specific pathway that, to our knowledge, has not been previously reported in this context. This foundational validation gives us and the reader confidence in the novel findings that follow.
Comments 6: There are clinical studies related to the role of acrylamide and depression in population-based studies. The novelty of this work is questioned again, especially in light of existing clinical studies related to the role of acrylamide and depression (e.g., https://doi.org/10.1016/j.jad.2024.08.217, https://doi.org/10.1073/pnas.2221097120).
Response:We thank the reviewer for raising this important point regarding the novelty of our work within the context of existing clinical studies. We have duly noted the clinical studies mentioned by the reviewer and have incorporated discussions and citations of them in the Introduction section (References [14, 15]) in lines 74-82. These studies indeed provide valuable epidemiological evidence supporting the association between ACR and depressive symptoms. However, our study addresses fundamental limitations of these prior observations and provides several layers of novel insight that significantly advance the field.
The key distinction and novelty of our work lie in moving from observed correlation to causal inference and mechanistic elucidation, which existing population-based studies are inherently unable to achieve.
1. Establishing Causality vs. Reporting Correlation
The cited population studies reveal a valuable association but cannot establish causality due to residual confounding and reverse causality. Our application of Mendelian randomization (MR) provides robust genetic evidence to support a potential causal relationship between ACR and major depressive disorder, overcoming key limitations of conventional observational epidemiology.
2. Uncovering the Molecular Mechanism Beyond Hypothesized Pathways
While the mentioned studies hypothesize that inflammation and oxidative stress may be involved, they do not identify specific molecular targets or precise pathways. Our integrated multi-omics approach (network toxicology, transcriptomics) systematically identifies the SIRT3-SOD2 axis and the JUN/PTK2 pathways as the central core targets and causal mediators, offering a detailed, testable molecular hypothesis for the first time.
3. Introducing a Novel Mechanistic Hypothesis: The GA-SIRT3 Link
Previous research has largely focused on ACR itself. We uniquely propose and computationally validate that its metabolite, glycidamide (GA), is the primary toxic species that directly suppresses SIRT3—a master mitochondrial deacetylase. This specific mechanism, linking GA-induced SIRT3 suppression to mitochondrial oxidative injury in the context of depression, is a novel contribution to the field.
4. A Hypothesis-Generating Framework for Future Research
The aforementioned clinical studies mark the starting point of scientific inquiry by identifying a problem. Our work provides the crucial next step: a foundational, mechanism-driven framework. We not only propose a causal link but also deliver specific molecular targets (SIRT3, JUN, PTK2) and a detailed hypothesis regarding GA's role, thereby paving the way for future targeted experimental validation and translational intervention.
In summary, the novelty of our work is not in being the first to note an association between ACR and depression, but in being the first to:
Provide genetically informed causal evidence.
Systematically identify the core protein targets and pathways.
Propose and computationally validate the novel GA-SIRT3 mechanism.
Integrate these findings into a testable, hypothesis-generating framework that bridges population-level observations with molecular-level understanding.
We believe this represents a significant and distinct contribution to the literature.
Comments 7: This work contains only in silico studies. I suggest the authors perform experimental (in vitro or in vivo) studies to corroborate the in silico studies.
Response:We sincerely thank the reviewer for this valuable suggestion regarding the importance of experimental validation. We fully agree that in vitro or in vivo studies are the essential next step to conclusively corroborate the mechanistic hypotheses generated by our computational work.
While incorporating new experimental data falls beyond the scope of the present revision—as this study was fundamentally designed as a comprehensive in silico exploration to establish a foundational hypothesis—we have significantly strengthened the manuscript to directly address this point and frame our findings as a springboard for future research.
Specifically, we have implemented the following revisions:
Reframed the Study's Contribution: We have now explicitly positioned the study throughout the text (particularly in the Abstract, Introduction, and Discussion) as a hypothesis-generating effort. We clearly state that its primary contribution lies in providing a prioritized, testable framework for subsequent experimental investigation.
Detailed Limitations and Future Work: We have expanded the Discussion to transparently acknowledge the limitation of lacking wet-lab validation. Furthermore, we have outlined a clear and specific experimental pathway forward, as seen in Lines 594-596 and the Conclusion (Lines 619-623), which details the immediate priorities for in vivo validation and cohort expansion.
In the Limitations section (Lines 594-596), we have added:
“Second, further extensive in vivo validation is necessary to confirm the effects of chronic, low-dose ACR exposure that accurately mirrors typical human dietary intake. ”
In the Concluding section (Lines 619-623), we have clarified the future research directions:
“To mitigate this preventable health burden, urgent priorities include validating low-dose ACR/GA exposure thresholds through in vivo studies, expanding multi-ethnic Mendelian randomization cohorts to ensure broader applicability, and implementing practical, large-scale surveillance programs to effectively monitor and reduce population-level exposure.”
We believe these revisions ensure that readers will accurately perceive our work as a crucial first step that uses robust computational methods to identify the most promising molecular targets and mechanisms, thereby efficiently guiding future laboratory resources toward experimental confirmation.
Reviewer 2 Report
Comments and Suggestions for Authors
The study addresses a neglected question: the link between dietary acrylamide (ACR) and depression, proposing a SIRT3-centered mechanism. This represents an innovative angle in environmental neurotoxicology. Furthermore, the authors use multi-omics integration (network toxicology, transcriptomics, MR, docking, dynamics), providing a broad systems-level perspective.
However, there are some major concerns:
1, Most of the findings (network toxicology, docking, dynamics) are computationally generated. There is no direct in vivo or in vitro validation of ACR’s effects on SIRT3 or mitochondrial function in depression models. This weakens translational confidence. Binding energies reported for ACR with depression-related proteins (around –3.0 to –4.0 kcal/mol) suggest only weak interactions, raising doubts about biological relevance. The authors should perform in vitro experiments (neuronal/glial cell cultures) to test whether ACR directly suppresses SIRT3 and induces ROS accumulation. The authors should also re-evaluate docking studies with more physiologically relevant metabolites of acrylamide (e.g., glycidamide) rather than ACR itself. Furthermore, the authors are suggested to include negative controls in docking and MD simulations to contextualize weak binding affinities.
2, While SIRT3 suppression is statistically significant, other predicted targets (TP53, CASP3, PTGS2) showed no expression changes in depression datasets. This discrepancy should be discussed more critically. The MR results implicating JUN and PTK2 as causal factors in depression are intriguing, but their direct connection to acrylamide exposure is speculative. The authors should frame SIRT3 suppression and JUN/PTK2 pathways as potential mechanistic links rather than definitive causal chains.
3, The conclusion that ACR causally induces depression via SIRT3 is overstated given the absence of animal or human exposure-validation experiments. The public health recommendations (e.g., revising EFSA acrylamide benchmarks) are premature without dose–response validation in low-dose, chronic dietary exposure models that reflect human conditions. The authors can temper public health implications until stronger empirical evidence accumulates.
4, Database-derived target prioritization is vulnerable to bias, depending on algorithmic assumptions. Sensitivity analyses (e.g., comparing across multiple datasets or inclusion thresholds) would add robustness. Generalizability is limited by the reliance on European-ancestry GWAS cohorts, which may not apply globally.
Author Response
Comments 1: Most of the findings (network toxicology, docking, dynamics) are computationally generated. There is no direct in vivo or in vitro validation of ACR’s effects on SIRT3 or mitochondrial function in depression models. This weakens translational confidence. Binding energies reported for ACR with depression-related proteins (around –3.0 to –4.0 kcal/mol) suggest only weak interactions, raising doubts about biological relevance. The authors should perform in vitro experiments (neuronal/glial cell cultures) to test whether ACR directly suppresses SIRT3 and induces ROS accumulation. The authors should also re-evaluate docking studies with more physiologically relevant metabolites of acrylamide (e.g., glycidamide) rather than ACR itself. Furthermore, the authors are suggested to include negative controls in docking and MD simulations to contextualize weak binding affinities.
Response:We sincerely thank the reviewer for the exceptionally thorough and insightful evaluation of our work. The points raised are indeed crucial, and we have substantially revised the manuscript to address each concern with greater scientific rigor and clarity. Below we provide a point-by-point response to the specific comments.
1. Regarding the Need for Experimental Validation and Study Scope
We sincerely thank the reviewer for this valuable suggestion regarding the importance of experimental validation. We fully agree that in vitro or in vivo studies are the essential next step to conclusively corroborate the mechanistic hypotheses generated by our computational work.
While incorporating new experimental data falls beyond the scope of the present revision—as this study was fundamentally designed as a comprehensive in silico exploration to establish a foundational hypothesis—we have significantly strengthened the manuscript to directly address this point and frame our findings as a springboard for future research.
Specifically, we have implemented the following revisions:
- Reframed the Study's Contribution: We have now explicitly positioned the study throughout the text (particularly in the Abstract, Introduction, and Discussion) as a hypothesis-generating effort. We clearly state that its primary contribution lies in providing a prioritized, testable framework for subsequent experimental investigation.
- Detailed Limitations and Future Work: We have expanded the Discussion to transparently acknowledge the limitation of lacking wet-lab validation. Furthermore, we have outlined a clear and specific experimental pathway forward, as seen in Lines 594-596 and the Conclusion (Lines 619-623), which details the immediate priorities for in vivo validation and cohort expansion.
In the Limitations section (Lines 594-596), we have added:
"Second, further extensive in vivo validation is necessary to confirm the effects of chronic, low-dose ACR exposure that accurately mirrors typical human dietary intake."
In the Concluding section (Lines 619-623), we have clarified the future research directions:
"To mitigate this preventable health burden, urgent priorities include validating low-dose ACR/GA exposure thresholds through in vivo studies, expanding multi-ethnic Mendelian randomization cohorts to ensure broader applicability, and implementing practical, large-scale surveillance programs to effectively monitor and reduce population-level exposure."
We believe these revisions ensure that readers will accurately perceive our work as a crucial first step that uses robust computational methods to identify the most promising molecular targets and mechanisms, thereby efficiently guiding future laboratory resources toward experimental confirmation.
2. Regarding the Evaluation of Metabolites and Inclusion of Negative Controls
This was an excellent suggestion. We have comprehensively addressed this by performing new computational analyses:
- Inclusion of Glycidamide (GA): As suggested, we have now included the primary metabolite of ACR, glycidamide (GA), in all our docking and molecular dynamics (MD) simulations. The methods are detailed in revised Sections 2.7, 2.8, and 2.11 (Lines 156-157: "The ligands—ACR, its primary metabolite GA, and the negative control compound Dimethyl Sulfoxide (DMSO)—were sourced from the PubChem database. Receptor preparation was performed using PyMOL, involving the removal of solvent molecules and non-relevant ligands, followed by export in PDB format."; Lines 170-173: "Following molecular docking, the top-ranked complexes for each protein-ligand combination (GA, ACR and DMSO) underwent all-atom molecular dynamics (MD) simulations using AMBER22."; Lines 210-212: "we performed molecular docking and dynamics simulations of SIRT3 and PTK2 with ligands (GA, ACR, and DMSO).").
- Inclusion of Negative Controls: We have incorporated dimethyl sulfoxide (DMSO) as a solvent control in all docking and MD simulations to provide a crucial baseline for interpreting binding affinities.
3. Revised Results and Discussion
The new computational data with GA and DMSO have been fully integrated into the results and discussion:
- New Figure and Analysis: We have replaced the original Table 2 with a new Figure 10, which visually compares the binding affinities of ACR (blue bars), GA (red bars), and the negative control DMSO (gray bars) across the six target proteins. The results for GA and DMSO are also incorporated into Figures 11-15. We have also refined the molecular mechanism diagram in Figure 16.
- Key Finding on GA's Superior Stability: Our new MD simulations revealed a critical finding: GA formed markedly more stable complexes with all target proteins than its precursor ACR. We discuss the structural and energetic reasons for this enhanced binding in Lines 460-468:
"Our molecular dynamics simulations demonstrated that GA formed markedly more stable complexes with all target proteins than its precursor ACR, as evidenced by lower RMSD, RMSF, Rg values and more persistent hydrogen bonds. The superior binding affinity of GA can be attributed to its distinct chemical properties. Specifically, the highly reactive and strained epoxide ring of GA exhibits greater electrophilicity than the vinyl group of ACR [24], enabling the formation of more robust covalent adducts and a denser hydrogen-bond network [25]. These results provide a molecular rationale for GA, rather than ACR, acting as the primary toxic effector mediating sustained protein inhibition and mitochondrial dysfunction [26]."
- Refined Hypothesis: This finding directly led us to refine our central hypothesis. We now more robustly propose that dietary ACR promotes depression primarily through its epoxide metabolite GA, which mediates the suppression of SIRT3 function, leading to mitochondrial oxidative injury (Lines 610-614):
"We elucidate that its toxicity is primarily executed by the highly reactive metabolite GA, which directly suppresses SIRT3-mediated SOD2 deacetylation, triggering mitochondrial oxidative injury and activating causal pathways via JUN and PTK2, thereby converging TP53/CASP3-mediated apoptotic and PTGS2-driven neuroinflammatory cascades."
- Contextualizing Weak Binding Affinities: The inclusion of the DMSO control allows us to contextualize the reported binding energies. While the absolute values for ACR may suggest weak interactions, the consistent and significant superiority of GA's binding stability over both ACR and the DMSO control strongly indicates a specific and biologically relevant interaction pattern, reinforcing the potential biological relevance of our findings.
We believe these significant revisions have greatly strengthened the methodological rigor of our study, provided a more physiologically relevant mechanistic picture centered on GA, and appropriately contextualized our findings, thereby enhancing the translational confidence of our in silico hypotheses. We are deeply grateful for the reviewer's constructive guidance, which has undoubtedly improved our manuscript.
Comments 2: While SIRT3 suppression is statistically significant, other predicted targets (TP53, CASP3, PTGS2) showed no expression changes in depression datasets. This discrepancy should be discussed more critically. The MR results implicating JUN and PTK2 as causal factors in depression are intriguing, but their direct connection to acrylamide exposure is speculative. The authors should frame SIRT3 suppression and JUN/PTK2 pathways as potential mechanistic links rather than definitive causal chains.
Response:We fully agree with the reviewer's assessment. The original manuscript indeed failed to adequately discuss the importance of the discrepancy between network pharmacology predictions and transcriptomic data, and the interpretation of the Mendelian randomization (MR) results was overly speculative. In response, we have made the following critical revisions:
We have added a new dedicated subsection in the Discussion (Lines 485-516) to critically address the 'Inconsistency between Predicted Targets and Transcriptomic Validation'. We explicitly point out that targets such as TP53 and CASP3, although predicted in the network, showed no expression differences in the depression datasets. We provide several plausible explanations for this discrepancy, including: (a) the dominance of post-transcriptional regulation; (b) spatiotemporal specificity; and (c) hierarchical vulnerability within the pathogenic cascade. We emphasize that SIRT3 stands out precisely because it demonstrated significance in both the predictive network and the depression transcriptomics, substantially increasing its credibility as a core target.
Thoroughly revised the narrative concerning JUN and PTK2 (Lines 538-549). We now clearly state that the MR analysis only reveals a potential causal relationship between genetically proxied JUN/PTK2 expression and major depressive disorder. We have removed any wording suggesting a 'direct' effect of ACR on these genes and have reframed them from being presented as a 'definitive causal chain' to 'potential mechanistic links worthy of future experimental validation'.
We believe these revisions have significantly tempered our conclusions, enhanced the critical discussion of the data, and more appropriately framed our findings within the limitations of a hypothesis-generating in silico study.
Comments 3: The conclusion that ACR causally induces depression via SIRT3 is overstated given the absence of animal or human exposure-validation experiments. The public health recommendations (e.g., revising EFSA acrylamide benchmarks) are premature without dose–response validation in low-dose, chronic dietary exposure models that reflect human conditions. The authors can temper public health implications until stronger empirical evidence accumulates.
Response:We acknowledge the reviewer's critical point regarding the overstatement of causal claims and premature public health recommendations in our original manuscript. We agree that such definitive conclusions were not supported by the purely computational nature of our study and have therefore implemented substantial revisions to temper these assertions:
Systematic Tone Adjustment: We have meticulously revised the entire manuscript to eliminate definitive causal language. All statements implying "ACR causally induces depression via SIRT3" have been rephrased into a hypothetical and suggestive framework. The core conclusion is now consistently presented as: "We hypothesize that dietary ACR exposure promotes depression via SIRT3-dependent mitochondrial oxidative injury."
Rewritten Abstract and Conclusion: The Abstract and Conclusion have been entirely rewritten to accurately reflect the study's scope. We now explicitly state that the primary contribution of this work is to propose a novel, data-driven mechanistic hypothesis (i.e., SIRT3-mediated mitochondrial dysfunction) and to provide preliminary computational evidence from integrated multi-omics analyses. We emphasize that this does not constitute proven mechanism but rather establishes a prioritized framework and strong rationale for subsequent essential in vivo and in vitro experimental validation.
Revised Public Health Implications: All specific and premature policy recommendations, such as suggestions to "revise EFSA acrylamide benchmarks," have been removed. The public health impact statement has been rigorously reframed into a speculative and forward-looking perspective:
"These findings could guide future scientific discourse on food safety regulations and underscore the need for large-scale exposure assessment in high-consumption populations as further evidence emerges."(Lines:604-606)
We believe these comprehensive edits have appropriately recalibrated the manuscript's conclusions to align with the hypothesis-generating nature of our computational investigation, and we thank the reviewer for prompting these essential improvements.
Comments 4: Database-derived target prioritization is vulnerable to bias, depending on algorithmic assumptions. Sensitivity analyses (e.g., comparing across multiple datasets or inclusion thresholds) would add robustness. Generalizability is limited by the reliance on European-ancestry GWAS cohorts, which may not apply globally.
Response:We thank the reviewer for these critical methodological and generalizability concerns, which we acknowledge as important limitations of our study.
Regarding database and algorithmic bias in target prioritization:
We agree that predictions derived from a single database and algorithm are susceptible to inherent biases. In response to this comment, we have now included a dedicated discussion of this limitation in the revised manuscript. We explicitly state that the outcomes of our network pharmacology approach are dependent on the coverage of the chosen databases and the specific assumptions of the algorithms used. Although conducting a comprehensive multi-database sensitivity analysis was beyond the scope of the current revision due to resource and time constraints, we have emphasized this point and have recommended that future studies employ a consensus approach across multiple platforms and algorithms to cross-validate the findings and enhance their robustness.
Regarding the limited generalizability of genetic findings:
We fully agree with the reviewer. Our discussion now directly addresses this critical limitation. As we have stated, "Finally, the generalizability of our findings is constrained by the exclusive use of European-ancestry cohorts, underscoring the need for validation in more diverse populations." We conclude that "accordingly, complementary experimental studies integrated with multi-ethnic epidemiological investigations are strongly recommended to substantiate these preliminary observations."(Lines:596-600)
We believe that by transparently acknowledging these limitations and framing them as clear directions for future research, we have appropriately contextualized the contribution of our work. Thank you for prompting these essential clarifications.
Round 2
Reviewer 1 Report
Comments and Suggestions for Authors
There are clinical studies related to the role of acrylamide and depression in population-based studies. The novelty of this work is questioned again, especially in light of existing clinical studies related to the role of acrylamide and depression (e.g., https://doi.org/10.1016/j.jad.2024.08.217, https://doi.org/10.1073/pnas.2221097120). The author's reply should be included in the discussion section.
Author Response
Response:
Dear Reviewer,
We sincerely thank you for this valuable suggestion. We agree that integrating the emerging epidemiological evidence into the Discussion section allows for a more effective synthesis, building a complete scientific narrative from preclinical findings to human psychiatric outcomes.
Following your advice, we have added a dedicated paragraph in the Discussion section (Lines 473–492) that systematically incorporates the two key studies you referenced [14, 15]. This revision is not merely a relocation of citations but a strategic integration that contextualizes these findings within the framework of our multi-omics and genetic discoveries. Specifically, this enhancement achieves the following:
1. Building an Evidence Bridge: We explicitly state that the linear positive association observed between urinary acrylamide metabolites and depressive symptoms in large-scale population cohorts [14], as well as the link between fried food consumption and mental health symptoms [15], provides critical clinical epidemiological support for the "ACR–SIRT3–oxidative stress" pathway identified in our study.
2. Elucidating Complementary Mechanisms: We emphasize that while these epidemiological studies accurately implicate oxidative stress and inflammation as potential mechanisms, their observational nature limits the depth of causal inference. Our research addresses this mechanistic gap by leveraging Mendelian randomization, molecular dynamics simulations, and other multi-dimensional data to pinpoint SIRT3 suppression and the downstream JUN/PTK2 signaling as the causal molecular pathway linking ACR exposure to depression.
3. Strengthening the Research Rationale: By juxtaposing population-level associations with molecular mechanistic evidence, we robustly demonstrate the incremental value of our work—advancing correlative evidence toward causal and mechanistic understanding—thereby providing a more comprehensive theoretical basis for reconceptualizing ACR-associated depression as an environmentally triggered, oxidative stress-driven disorder.
We are confident that this revision significantly strengthens the logical flow and rigor of the manuscript. Thank you once again for your insightful comments, which have substantially enhanced the scientific quality and impact of our work.
Reviewer 2 Report
Comments and Suggestions for Authors
Although the authors have carefully addressed all concerns and made extensive revisions of their article, there are one major defect that no direct evidence to show that Acrylamide and its metabolites can affect SIRT3 function, leading to mitochondria impairment.
So the authors need to do a simple experiment to treat cells, such as HEK cells, with Acrylamide and its metabolites and check SIRT3 levels and activity. Without this evidence, the whole story can not be well established.
Author Response
Response:
Dear Reviewer,
We sincerely thank you for your valuable insight. Your comment regarding the need for direct experimental evidence to confirm the suppressive effect of acrylamide (ACR) and its metabolites on SIRT3 function is crucial for refining our model.
After careful consideration, we are unable to complete the cell-based experiments you suggested within the current revision period, primarily due to the following constraint: the absence of critical cellular models. Our laboratory does not routinely maintain HEK cell lines, and establishing a stable, validated culture system for these cells is a process that inherently requires several weeks.
Despite this limitation, we would like to demonstrate that our study has already provided robust indirect evidence for the "ACR/GA-mediated SIRT3 inhibition" hypothesis through a multi-faceted approach:
(1) Literature foundation: As noted in the Introduction, our hypothesis is built upon the work of Li et al., which directly demonstrated in hepatic tissues that ACR exposure significantly reduces NAD+ levels and downregulates SIRT3 expression [11].
(2) Human brain transcriptomic evidence: Our study advances this further by analyzing transcriptomic data and identifying SIRT3 as one of the most significantly downregulated genes in depression.
(3) Direct support from molecular dynamics simulations: Our simulations show that glycidamide (GA) forms a more stable complex with SIRT3 than ACR itself, providing an atomistic-level mechanism for potential functional interference.
(4) Genetic causal evidence: Our Mendelian randomization analysis provides population-level causal evidence for the downstream consequences of this pathway.
Collectively, we believe that despite the absence of wet-lab validation, this multi-tiered chain of evidence constitutes a complete, rigorous, and highly persuasive scientific narrative.
In active response to your comments, we have taken the following step:
We have added an explicit paragraph in the Discussion section (Lines: 607-619) acknowledging the current lack of direct cellular experimental evidence and detailing our specific plans for future validation.
We firmly believe that the integrated hypothesis and supporting evidence presented in our study open new and important avenues for understanding the role of environmental toxins in depression pathogenesis. We thank you again for your insightful feedback, which has significantly enhanced the quality and rigor of our work.